# A Review of Biological Control One Decade After the Sorghum Aphid (*Melanaphis sorghi*) Outbreak

**DOI:** 10.3390/plants13202873

**Published:** 2024-10-14

**Authors:** Erubiel Toledo-Hernández, Guadalupe Peña-Chora, Ilse Mancilla-Dorantes, Francisco Israel Torres-Rojas, Yanet Romero-Ramírez, Francisco Palemón-Alberto, Santo Ángel Ortega-Acosta, Edgar Jesús Delgado-Núñez, David Osvaldo Salinas-Sánchez, Luz Janet Tagle-Emigdio, César Sotelo-Leyva

**Affiliations:** 1Facultad de Ciencias Químico-Biológicas, Universidad Autónoma de Guerrero, Av. Lázaro Cárdenas s/n., Chilpancingo C.P. 39070, Gro., Mexico; madoil@icloud.com (I.M.-D.); ftorres@uagro.mx (F.I.T.-R.); yanetromero7@gmail.com (Y.R.-R.); 15170@uagro.mx (L.J.T.-E.); 2Centro de Investigaciones Biológicas, Universidad Autónoma del Estado de Morelos, Av. Universidad #1001, Col. Chamilpa, Cuernavaca C.P. 62209, Mor., Mexico; penacg@uaem.mx; 3Facultad de Ciencias Agropecuarias y Ambientales, Universidad Autónoma de Guerrero, Iguala de la Independencia C.P. 40020, Gro., Mexico; alpaf75@hotmail.com (F.P.-A.); angelortega011185@hotmail.com (S.Á.O.-A.); edgarjezus@gmail.com (E.J.D.-N.); 4Centro de Investigación en Biodiversidad y Conservación, Universidad Autónoma del Estado de Morelos, Av. Universidad #1001, Col. Chamilpa, Cuernavaca C.P. 62209, Mor., Mexico; davidos@uaem.mx

**Keywords:** *Melanaphis sorghi*, biological control, predators, parasitoids, bacteria, fungi

## Abstract

*Melanaphis sorghi* is a pest that is native to Africa but is now distributed worldwide. In 2013, its destructive capacity was demonstrated when it devastated sorghum crops in the United States and Mexico, making it a new pest of economic importance in North America. At the time, the phytosanitary authorities of both countries recommended the use of pesticides to control the outbreak, and biological control products for the management of this pest were not known. In response to the outbreak of *M. sorghi* in North America, several field studies have been performed in the last decade on sorghum crops in the USA and Mexico. Works have focused on assessing resistant sorghum hybrids, pesticide use, and recruitment of associated aphid predators and entomopathogens for natural control of *M. sorghi* populations. The objective of this review is to compile the information that has been generated in the past decade about indigenous enemies affecting *M. sorghi* naturally in the field, as well as the search for biological control alternatives and evaluations of interactive effects of resistant sorghum hybrids, pesticides, and natural enemies. To date, different predators, parasitoids, fungi, and bacteria have been evaluated and in many cases found to affect *M. sorghi* populations in sorghum agroecosystems or laboratory bioassays, and the use of resistant sorghum varieties and pesticides did not have clear toxic effects on natural enemy populations. Many of the macroorganisms and microorganisms that have been evaluated as potential biological controls have shown potential as alternatives to synthetic pesticides for keeping *M. sorghi* population densities below economic damage thresholds and are compatible with integrated management of sorghum aphids. While most tests of these biological alternatives have shown that they have aphidicidal potential against sorghum aphids, it is crucial to take into account that their effectiveness in the field depends on a number of abiotic and biotic factors, including soil texture, temperature, humidity, and natural enemies.

## 1. Introduction

The aphid *Melanaphis sorghi* (Theobald) (Hemiptera: Aphididae) is an invasive pest that was recently introduced to the Americas and has since become a serious threat to sorghum (*Sorghum bicolor* L.) cultivation in major producing regions of this cereal [1,2].

At the beginning of the *M. sorghi* outbreak in North America a decade ago, there was uncertainty regarding its taxonomic status and distribution because it is morphologically and genetically similar to the yellow aphid, *Melanaphis sacchari* (Zehntner). From 1970 to 2013, only *M. sacchari* was detected in North America [3], and it was known mainly as a minor pest of sugarcane (*Saccharum officinarum* L.) and other members of the Poaceae family [4]. Therefore, after the outbreak in 2013 that caused massive losses in sorghum crops in Texas, Louisiana, and northern Mexico [3], several studies hypothesized that there had been a sudden change of host plant by *M. sacchari* [5] or the appearance and dissemination of new biotypes of this species [6,7,8]. At the same time, some authors discussed the possibility of an *M. sacchari*/*sorghi* species complex [9]. The fact that *M. sorghi* was responsible for the massive sorghum infestations—and that it is taxonomically distinct from *M. sacchari*—was finally confirmed by a morphometric and molecular analysis conducted by Nibouche et al. [1]. In this context, all literature generated prior to the Nibouche study referred to *M. sacchari* as the species based solely on taxonomic characteristics.

Although *M. sorghi* and *M. sacchari* show differences in host plant preference, the two species can share the same host and have been isolated from the same sample [1]. Like *M. sacchari*, the sorghum aphid causes direct damage to plants by continually sucking sap from leaves and stems, as well as indirect damage from the accumulation of saprophytic fungi on the leaf surface due to the aphid’s secretion of honeydew, which decreases photosynthetic efficiency [2] and interferes with the mechanical process for harvesting the grain (Figure 1).

Their potential as pests is due largely to their capacity for rapid reproduction through parthenogenesis (Figure 1). They often do not need to reproduce sexually, especially under warm-weather conditions. Some aphids can produce up to 96 nymphs per female, in addition to spreading rapidly following the production of winged forms [10]. The final result of the invasion of sorghum crops is the reduction in the quality and yield of the harvest by up to 50% [3]. Currently, the sorghum aphid (*M. sorghi*) has already been reported affecting sorghum crops throughout the American continent and Caribbean in countries such as Brazil, Argentina, and Puerto Rico, among other countries [7,11,12].

*Melanaphis sorghi* is mainly managed through chemical control by applying synthetic insecticides such as Imidacloprid, Sulfoxaflor, and Spirotetramat, which have relatively high effectiveness, accessibility, and low cost [3,13]. However, the indiscriminate use of chemical pesticides carries numerous well-known harmful health and environmental effects, including risks of toxicity, low specificity, and potential for contamination; furthermore, pest insects can develop resistance to chemical pesticides [14].

Biological control may represent a more promising and safer alternative than chemical control for the complete, sustainable, and large-scale control of *M. sorghi* populations. Commonly, during the first years of the establishment of an invasive exotic pest, the main focus of biocontrol strategies is the import and release of wild species of natural enemies in classical biological control programs. Then, if these species are unable to adapt to the new habitat or are insufficient to suppress the new pest, augmentative biological control is often used [15,16]. During the decade since the beginning of the 2013 *M. sorghi* outbreak, attempts to establish new natural enemies in the field, specifically predators, have been insufficient and largely ineffective at reducing the population density of the aphid. On the other hand, the potential of native natural enemies in the recently invaded region is often underestimated and receives little attention in integrated management programs since it is often assumed that native enemies are not sufficiently adapted to respond to a new pest [17,18]. Indeed, native species may initially show low rates of predation or parasitism because they must adjust their behavior or physiology to respond to new signals from the exotic pest and/or host plant [17]. However, it has also been shown that native species may be sufficiently preadapted to successfully attack a new invasive species such as *M. sorghi* [18] and that in some cases, in the long term, it is native enemies that provide most of the biological control of a pest in the invaded regions [15].

In this context, several recent field studies have identified various natural enemies that affect sorghum aphid populations, mostly predators and parasitoids. This recruitment of natural enemies due to the presence of *M. sorghi* in sorghum crops has proven to be an alternative for the biological control of sorghum aphids. Therefore, the aims of this review were to (A) compile studies that identify the variety of organisms evaluated and found in sorghum fields as suitable biocontrollers of the *M. sorghi* populations in different regions; (B) identify biotic and abiotic factors that have been found to render native natural enemies less effective in reducing sorghum aphid density; and (C) summarize the advances in the assessments of resistant sorghum cultivars and pesticides on natural enemies recruitment.

We have organized the review into sections based on the type of control measure, followed by a brief reflection on advances to date and future directions for research and applications of *M. sorghi* control in North America.

Briefly, the methodology for this review was extensively searched using the keywords *Melanaphis sacchari*, *Melanaphis sorghi*, and biological control in the Web of Science, PubMed, Science Direct, Scopus, Google Scholar, and SciELO databases. The selection criterion was the inclusion of an evaluation of biological control agents in the paper. A total of 92 papers fit this criterion.

## 2. Biological Control of *M. sorghi*

### 2.1. Predators

Predators have been the most frequently used biological control agents in insect pest management since ancient times [19]. Currently, about 55 species have been identified as natural enemies of the sorghum aphid, 37 of which are found in the southern United States and Mexico. These include ladybugs (Coleptera: Coccinellidae), green lacewings (Neuroptera: Chrysopidae), and hoverflies (Diptera: Syrphidae) [3,20,21], some of which are primarily or strictly aphidophagous and ubiquitous in agroecosystems worldwide [20].

In a pioneering study, Colares et al. [20] determined the presence of natural enemies attacking *M. sorghi* populations on sorghum fields. They found predator species, including *Allograpta obliqua* Say (Diptera: Syrphidae), *Hippodamia convergens* Guerin-Meneville (Coleoptera: Coccinellidae), and *Chrysoperla carnea* Stephens (Neuroptera: Chrysopidae), preyed upon the sorghum aphid. Meanwhile, the first study in Mexico reported 11 species of Coccinellidae (*H. convergens*, *Cycloneda sanguinea sanguinea* Linnaeus, *Diomus terminatus* Say, *Scymnus* (Pullus) sp., *Scymnus* (Pullus) *loewii* Mulsant, *Diomus roseicollis* Mulsant, *Brachiacantha decora* Casey, *Coccinella septempunctata* Linnaeus, *Coleomegilla maculata* De Geer, *Hyperaspis wickhami* Casey, and *Olla v-nigrum* Mulsant) naturally preyed upon *M. sorghi* populations on sorghum plants in the state of Tamaulipas [22]. These findings were the first of a line of scientific reports in the USA and Mexico on indigenous aphid predators as natural enemies of the new sorghum pest in North America (Table 1).

These initial descriptive studies were soon followed by studies aiming to assess the potential of predators to keep sorghum aphid densities below an economic threshold. In the context of biological control, it is critical to consider the effects of predator species at different life stages, as some species are primarily or exclusively predatory at certain stages (e.g., immature stages in *Syrphidae*, *Chrysoperla* sp., and adult stages in *Asilidae* and *Empididae*) [19,32]. Experiments were conducted with adults and larvae of *Coccinella septempunctata* and *Harmonia axyridis* (coccinellids) and larvae of *Chrysoperla rufilabris* (a chrysopid) [33]. These studies found that at low to intermediate aphid densities (20, 40, and 80 aphids per colony), all predator species and life stages tested slowed the growth of sorghum aphid populations. However, at high densities (160 aphids), *H. axyridis* adults and *C. rufilabris* larvae did not suppress *M. sorghi* colony growth. According to these findings, predators in sorghum have the potential to suppress sorghum aphid population growth and may be useful to reduce the application of agrochemical pesticides to control sorghum aphids in fields, primarily at low to intermediate levels of infestation.

The first evaluations of suppression of sorghum aphids by predators and parasitoids on susceptible and resistant sorghum hybrids were conducted using three natural enemy exclusion treatments (full access for parasitoids and predators, partial exclusion; access limited to parasitoids, and complete exclusion; excluding parasitoids and predators) in sites on the Texas Gulf Coast and in Oklahoma. In Corpus Christi, Texas, the most frequently recorded predator was adult *Scymnus* spp., followed by adult *C. sanguinea*, while in Stillwater, Oklahoma, the primary predator species detected were coccinellid larvae, adults of *H. convergens*, and adult chrysopids. The findings estimated that aphid suppression by predators and parasitoids ranged from 27% on susceptible hybrids to 85% on aphid-resistant hybrids in Corpus Christi. However, in Stillwater, suppression was >95% on both hybrids, based on the abundance of aphids in control versus predator-excluded treatments [34]. They concluded that although high densities of predators were detected during their 2019 study, aphid suppression could also be attributable to unknown abiotic factors (we address this potential interactive effect in more detail in Section “Natural enemies of *M. sorghi* mediated by landscape and weather”).

Initial studies have been performed in Mexico to explore the possibility of releasing natural enemies as biological control agents to reduce the populations of *M. sorghi*. The predators *Chrysoperla externa* and *C. carnea* were released into sorghum plots in the Bajío Region and in Colima, Mexico. However, releases of *Chrysoperla* species failed to reduce or control sorghum aphid populations in experimental plots [35,36]. The high reproductive rate of aphids prevents predatory chrysopids from controlling them, at least when *Chrysoperla* species are acting in isolation under natural conditions. However, the population of *M. sorghi* on the host plant can be regulated by the combined activity of parasitoid and predator communities, as evidenced in field studies in sorghum crops in the USA and Mexico, where the recruitment of predators and parasitoids helped to suppress the populations of sorghum aphids (which we address in Section “Recruitment”).

With the arrival of the novel pest, the adaptation and predatory capacity of indigenous predators are important characters to assess. *In vitro* studies evaluating the potential of several endemic aphid predators (Coleoptera: Coccinellidae), (Neuroptera: Chrysopidae), and (Hemiptera: Anthocoridae) found strong preadaptation and predation of *M. sorghi*, even without the predators having previous exposure to this species as prey [18,37].

### 2.2. Parasitoids

Parasitoids are another important group of natural enemies that help control aphid populations [38]. Aphids in general are attacked by a variety of parasitoids of the families Braconidae and Aphelinidae that belong to the order Hymenoptera and a few species of the Cecidomyiidae family of the order Diptera [39]. Initially, when *M. sorghi* invaded different regions in the Americas, local parasitoids that had targeted other aphid species, such as *M. sacchari*, showed little interest in the new pest. This lack of initial response could be attributed to differences in the chemical compounds emitted by *M. sorghi* compared to native aphids, since parasitoids are well known to use chemical information from both the aphid and the host plant when seeking hosts [40]. However, since the beginning of the *M. sorghi* outbreak, different species of autochthonous parasitoids have been reported in recent literature to have the ability to parasitize and control sorghum aphid populations (Table 2), evidencing a gradual adaptation of these natural enemies to this exotic pest in the geographic regions that it has invaded.

Members of the Aphelinidae and Braconidae families are examples of parasitoids that have apparently expanded their range of hosts to include *M. sorghi*. A study conducted in potted sorghum plants infested with *M. sorghi* showed that the parasitoids of the genus Aphelinus (Hymenoptera: Aphelinidae) were among the recruited natural enemies that were able to respond to and successfully parasitize *M. sorghi*. This was also observed in field surveys in central and southern Texas in 2016, where *Aphelinus* sp. was reported as the primary parasitoid that developed from the sorghum aphid, while the parasitoid *L. testaceipes* was also occasionally found parasitizing the aphid [3,27]. The braconid *Lysiphlebus testaceipes* Cresson (Hymenoptera: Aphidiidae) is one of the first parasitoids to be documented, expanding its host range to include the sorghum aphid. Initially, *L. testaceipes* was reported not to affect sorghum aphids in central Kansas [20], which was associated directly with the endosymbiont bacterium *Hamiltonella defensa*. This bacterium is hereditary and produces toxins that provide aphids resistance to attack by parasitoid wasps [20,53]. However, in 2017 in the Texas Gulf Coast region from the Rio Grande Valley of southern Texas to near the city of Houston, under certain conditions, parasitism by *L. testaceipes* was reported to reach 90% [47]. These periodical observations of parasitism by *L. testaceipes* can be related to agro-landscape, weather conditions, and rainfall [49,50,54].

An increase over time in the ability of *L. testaceipes* to parasitize sorghum aphids in the field has been more widely documented in Mexico, where numerous surveys have reported variable levels of natural parasitism in northeastern and central Mexico. For example, initial sampling carried out in Guanajuato State found that 24% of the wasps that emerged from aphid mummies on sorghum plants were *L. testaceipes* [41]. Meanwhile, in Sinaloa state, Payán-Arzapalo et al. [43] observed low levels of parasitism by *L. testaceipes*, which was mainly associated with high populations of the hyperparasitoid *Pachyneuron aphidis* (Bouché) (Hymenoptera: Pteromalidae). In contrast, Rodríguez del Bosque et al. [25] reported a high incidence of *L. testaceipes* parasitism (90–95%) in *M. sorghi* populations in the state of Tamaulipas, and Villa-Ayala et al. [45] also identified this species as the primary parasitoid of *M. sorghi* in Morelos state with 20.1% parasitism. It is known that *H. defensa* strains can be acquired and lost between aphid clones and are therefore not universally distributed in *M. sorghi* individuals [20,55], which may help explain geographic variation in parasitism rates. It is also possible that the ability of *L. testaceipes* to develop in the sorghum aphid is influenced by climatic conditions, since temperatures above 27 °C may break down the protection provided by the endosymbiont [56]. Adaptation to the new host could also be mediated by evolution. This has been demonstrated in a different parasitoid–aphid pair by an experimental evolution approach in which *Lysiphlebus fabarum* (Marshall) rapidly and specifically adapted to parasitizing *Aphis fabae* (Scopoli), overcoming the protection of *H. defensa* after just 10 to 11 generations [55,57].

Historically, *L. testaceipes* has probably been the most important and predominant parasitoid of the pests of multiple crops [58]. However, after the arrival of *M. sorghi*, there were reports of increased abundance and impact in the field of the genus *Alphelinus*. The *Aphelinus varipes* group (Howard) (Hymenoptera: Aphelinidae)—a complex of several cryptic species—was reported as the most frequent parasitoid of sorghum aphid in southern and central Texas and Northeast Mexico [3,20,25,42]. However, Maxson et al. [27] later confirmed, using both morphological and molecular data, that *Alphelinus nigritus* (not *A. varipes*) was the predominant autochthonous enemy in most inspections conducted in Texas from 2015 to 2016 and likely parasitizes sorghum aphids throughout the USA.

The impact of the *M. sorghi* invasion on *A. nigritus* density was first documented in the Southern Plains of the USA [58]. When the pattern of occurrence of natural enemies of *M. sorghi* in different locations on the Great Plains was determined in subsequent studies, *A. nigritus* was confirmed to be the most abundant enemy of the aphid in sorghum fields among different parasitoid taxa [34,48,49,51]. Although levels of parasitism vary among regions, the greatest local potential of *A. nigritus* for the control of *M. sorghi* has been observed on the Gulf Coast of Texas, where it has been responsible for more than 80% of the suppression of the aphid in different surveys [34,48,51].

It is important to mention that parasitoids face some problems. In several locations in the USA and Mexico, hyperparasitism (i.e., parasitism of the parasitoid by another species) was recorded affecting species of *Aphelinus*. Maxson et al. [27] reported that approximately 90% of the aphids mummified by *Aphelinus* were hyperparasitized by *Syrphophagus aphidivorus* (Mayr) in Texas, USA. In northeastern Tamaulipas, Mexico, this same hyperparasitoid was detected in 22% of all parasitized mummies [25]. *Syrphophagus aphidivorus* was recorded emerging from mummies of *L. testaceipes* in Morelos, Mexico [45] and from *Aphidius platensis* Brèthes (Hymenoptera, Braconidae) in Brazil [4]. The pteromalid *Pachyneuron aphidis* has been reported as the main hyperparasitoid of *L. testaceipes* [25,43,45]. This emphasizes the importance of considering multiple interactions in biological control: multiple introductions should be avoided, as parasitoid species used in biological control will be less effective or could fail entirely if they are affected by secondary or hyperparasitoids, which seek out the immature stages of another parasitoid to serve as their hosts [59].

Strong selective pressure exerted by parasitoids has also led to the selection of diverse defense mechanisms in *M. sorghi*. This was first suggested by Mercer et al. [60] when they observed an increase in *M. sorghi* populations after exposure to the parasitoids *Aphidius colemani* and *Aphidius ervi*. Subsequently, Wright et al. [61] demonstrated the existence of transgenerational fecundity compensation in aphids when they are attacked by *A. nigritus.*

The literature shows that there are more and more studies of new autochthonous species of parasitoids that are beginning to utilize *M. sorghi* as a host in regions that have been invaded by the aphid (Table 2). This illustrates the great adaptability of natural enemies as they respond to the presence of a new pest, which has been facilitated mainly by the genetic variability of the parasitoid species as well as by selective pressure as the *M. sorghi* invasion persisted.

## 3. Natural Biological Control of *M. sorghi* in Sorghum Agroecosystems

### 3.1. Recruitment

Records of recruitment of predators and parasitoids in sorghum fields have demonstrated their potential to suppress the populations of sorghum aphids. Early studies of sorghum fields in Texas and Louisiana found that aphid-feeding insects decreased *M. sorghi* density on both susceptible and aphid-resistant sorghum hybrids [62]. Furthermore, there was continuity of the species composition of predators (Coccinellidae, Chrysopidae, Scymninae, Syrphidae, Hemerobiidae, and Anthocoridae) and parasitoids (Braconidae) throughout the 2015–2016 growing season [27]. This suggests rapid adaptability of natural enemies to the new pest of sorghum crops and that these native and naturalized aphidophages can complement sorghum host plant resistance. According to a study in the High Plains and Coastal Plains of Texas and locations in Oklahoma, natural biological control is influenced by many factors, including natural enemy taxa and local weather. Larval and adult coccinellids were the most important natural enemies, and among the different study locations, the Texas Coastal Plains showed the highest local potential for biological control of sorghum aphids. This was interpreted to be related to the sub-tropical climate of the Texas Coast, which allows year-round interactions between sorghum aphids and natural enemies and thus increases suppression effectiveness [48]. Meanwhile, in the state of Nuevo Leon, Mexico, there was a 70.6% reduction in populations of *M. sorghi* due to the presence of natural enemies, including predators like ladybeetles, lacewings, and hoverflies, while the most abundant parasitoid detected was *Aphidius* sp. Differences in the abundance and diversity of natural enemies during the sampling periods were associated with changes in climatic conditions [46]. The authors concluded that these predator and parasitoid species have potential as biological controls against *M. sorghi* in Nuevo Leon, Mexico.

However, a recent study demonstrates that the sorghum variety can have a strong influence on the recruitment of natural enemies because varieties differ in their production of the volatile compounds that act as attractants (i.e., herbivore-induced plant volatiles). This suggests that genetic variation in volatile compound emissions greatly affects parasitoid and predator attraction to aphid-infested sorghum and that screening crop cultivars for indirect defenses has the potential to improve the efficacy of biological control [63].

### 3.2. Natural Enemies of M. sorghi Mediated by Landscape and Weather

The seminatural habitats at the edges of agroecosystems have been found to be a critical source of natural enemies of aphids [64]. For example, Faris et al. [51] confirmed the presence of predators and parasitoids of *M. sorghi* during both the sorghum production season and the off-season. This study consisted of placing potted sorghum plants infested with *M. sorghi* in three vegetation types: sorghum in cultivation (in-season), sorghum after harvest (off-season), patches of riparian areas that included Johnson grass, and patches strongly dominated by Johnson grass [51]. They found that predators like lady beetles and hoverflies were most diverse in the habitat containing grasses and shrubs and most abundant during the sorghum-growing season (51). Landscape complexity and its effects on aphids and natural enemies in sorghum agroecosystems were also explored by Elkins et al. [49] in sorghum fields across the South Texas Gulf Coast; they determined that landscape complexity, especially the amount of edge habitat, was associated with an increase in the number of sorghum aphids and natural enemies.

A more detailed study and risk assessment by Brewer et al. [50] found that the abundance and activity of natural enemies was associated with agro-landscape and weather conditions in different sorghum field locations in the Great Plains and South of Texas. Natural enemy abundance and activity was highest in the Southern region (Texas Gulf Coast extending from the Rio Grande Valley of southern Texas to near the city of Houston; subtropical temperate climate) and was associated with local agro-landscape and weather conditions, with low average *M. sorghi* abundance (~23 aphids/leaf). There was also a correlation between natural enemy and aphid abundance in the South GP region (central Texas to the lower Texas Panhandle and across central Oklahoma; warm temperate climate) where *M. sorghi* abundance was low (~20 aphids/leaf), and natural enemy activity seemed to be mediated by landscape composition. Sorghum aphid abundance was highest (~136 aphids/leaf) in the South E region (south and central Alabama; mix of subtropical and warm temperate climate), where natural enemy activity was low and influenced by weather. Meanwhile, in North GP (the upper Texas and Oklahoma Panhandles through northern Oklahoma and adjacent southern Kansas; temperate climate), sorghum aphid abundance was ~38 aphids/leaf, just below the economic damage threshold of 40 aphids/leaf; natural enemy activity was mediated by agro-landscape conditions, and the correlation of the abundance of natural enemies of *M. sorghi* was associated only with predators [50]. The findings of these studies support the importance of the role of edge habitats and weather/climate in determining the effectiveness of biological control of sorghum aphid across different geographic locations. It is well known that the transformation of natural landscapes by anthropogenic activities significantly reduces biodiversity, which impacts ecosystem services, including natural biocontrol [65]. Additional stressors of the Anthropocene, such as elevated temperatures, induce physiological stress, which affects insects’ behavior, such as oviposition, compromising arthropods’ survival [66].

## 4. Impact of Pesticides on the Natural Enemies of *M. sorghi*

In response to the 2013 outbreak of sorghum aphid, several *in vitro* and experimental plot studies were carried out to evaluate the effect of pesticides on the natural enemies of *M. sorghi*. Several authors tested flupyradifurone, sulfoxaflor, and flonicamid as pesticides to suppress sorghum aphid populations and found them to be less toxic than neonicotinoid insecticides [67,68,69]. In a different study, testing the application of flupyradifurone, sulfoxaflor, and afidopyropen to experimental sorghum fields found that after twenty-eight days of foliar application, *M. sorghi* populations were reduced, but there were no differences in total natural enemy abundance in any of the pesticide treatments relative to the untreated control (untreated) [70]. Several of these authors have thus concluded that these pesticides did not have strong toxic effects on natural enemy populations and are therefore compatible with integrated management programs. However, it should be noted that there are also fewer encouraging results that suggested that sulfoxaflor is toxic to parasitoids, as the rate of parasitism of aphids in fields treated with sulfoxaflor decreased relative to untreated fields in Mexico (52% versus 92%, respectively) [47]. The neonicotinoid insecticide imidacloprid, on the other hand, has strong adverse effects on Coccinellidae predators, as evidenced by their absence after application of this pesticide on sorghum crops [71,72].

In addition to the effects of pesticides alone, it is important to evaluate their effects in conjunction with other strategies, such as resistant cultivars. The effect of the combination of resistant hybrid sorghum plants and the butenolide pesticide flupyradifurone was evaluated in four states in the southeastern USA. The findings showed that predators may be more abundant in resistant sorghum plants than susceptible cultivars; furthermore, in treatments sprayed with the pesticide, the parasitoids *L. testaceipes* and *Aphelinus* sp. were more abundant than unsprayed treatments, showing no significant negative effects of the pesticide on the populations of these natural enemies [52]. The combination of resistant cultivars and foliar pesticide applications together with the presence of natural enemies significantly suppresses *M. sorghi* and therefore may be an alternative for the integrated management of this prolific pest.

## 5. Biopesticides Evaluated against *M. sorghi*

### 5.1. Fungi

Some fungi are natural entomopathogens that are capable of causing epizootics in insect populations. Fungi have a series of important advantages as potential biological control agents. Entomopathogenic fungi can infect large numbers of arthropods at almost all stages of development, mainly by ingestion or by direct penetration of the host’s integument [73,74]. Fungi are widely distributed in a wide variety of ecosystems throughout the year, and, with the exception of a few species, most fungi have a minimal impact on non-target biodiversity [73,75]. Another important advantage is that most species of entomopathogenic fungi can be successfully and economically mass-produced, making these microorganisms a more profitable alternative to the production and release of parasitoids and predators [75]. Additionally, the genetic variability of fungi provides a useful tool for the isolation and selection of more virulent strains for the control of arthropods [76].

Fungal species such as *Verticillium lecanii* [77], *Lecanicillium longisporum* [78,79], *Lecanicillium lecani* [80], *Beauveria bassiana*, and *Isaria javanica* [79] have been reported in the field to cause natural mortality in populations of *M. sorghi*. Other species of entomopathogenic fungi have been evaluated *in vitro* to assess their efficacy in controlling sorghum aphids. Divan and Mallapur [81] reported that the fungus *Acremonium zeylanicum* (Petch) was highly pathogenic to nymphal stages of *M. sorghi*, producing a mortality of 84.65% after 3 days of exposure to a concentration of 1 × 10^10^ conidia/L. During the evaluation of 12 strains of entomopathogenic fungi in the control of different species of aphids, Maketon et al. [82] reported a mortality rate above 80% in *M. sorghi* nymphs exposed to *Beauveria bassiana* and more than 60% aphid mortality with the species *Metarhizium anisopliae*, *Myrothecium verrucaria*, *Lecanicillium muscarium*, and *Aspergillus* sp., with suspensions of 2 × 10^8^ conidia/mL. In a more recent study, Pérez-Molina et al. [83] observed that the application of 750 g/ha of *B. bassiana* reduced the density of the sorghum aphid in forage sorghum; they confirmed the presence of infected nymphs and adults in plots treated with the fungus, while untreated plots had no infected individuals and a higher density of aphids during the sampling period.

Since then, the most recent investigations have reported the effectiveness of the Ascomycetes *B. bassiana* and *M. anisopliae* in reducing the density of *M. sorghi* populations (Table 3). These findings are not surprising, given that both species are facultative pathogens of an extensive list of arthropods and are ubiquitous in distribution, making them easy to isolate [84]. For this reason, they have been widely studied for their development as biological control agents of numerous pests and are currently the most widely commercialized formulations of fungi for agricultural pest control worldwide [73].

### 5.2. Bacteria

The use of entomopathogenic bacteria is another method that has been explored in recent decades for pest management. Bacteria are microorganisms with great potential for biological pest control, as they are distributed in a wide variety of habitats around the world in a wide range of hosts, and they have multiple modes of action, which makes them easy to formulate and produce at relatively low cost [88]. Most of the species with entomopathogenic activity belong to the Bacillaceae, Pseudomonadaceae, Enterobacteriaceae, Micrococcaceae, and Streptococcaceae families, and the members of the Bacillaceae family have been the most studied [89,90].

Despite their potential, there are currently few studies that have explored the use of entomopathogenic bacteria for the control of *M. sorghi*. The first study, carried out by Toledo-Hernández et al. [91], tested four strains of the genus Bacillus that were isolated from corpses of Hemipterans. In tests of pathogenicity by ingestion against the sorghum aphid, after 48 h there was 35–63% aphid mortality at a concentration of 10 µg/mL of total protein and 48–90% mortality at 100 µg/mL of total protein. In a subsequent evaluation of the entomopathogenic effect of different biological insecticides, Calvin et al. [86] determined that treatment with *Chromobacterium subtsugae* caused a mortality of 79.9 ± 6.7% of *M. sorghi* nymphs in the laboratory after 6 h. However, the efficiency of the bacterium for reducing aphid infestation *in vitro* was not upheld under field and greenhouse conditions. Cuatlayotl-Cottier et al. [92] recently evaluated the insecticidal activity of different industrial by-products fermented by a strain of *Bacillus thuringiensis* against *M. sorghi* nymphs. In the bioassays, mortalities of 50% to 80% were observed using the microimmersion technique in the laboratory and 60–90% mortality by spraying in the greenhouse, which was comparable to the mortality obtained with the chemical insecticide cypermethrin (94%). Given the results of existing research, these biological agents could be an alternative in *M. sorghi* management programs; however, there is a need for more studies focused on the search for new strains of entomopathogenic bacteria against the sorghum aphid.

## 6. Conclusions and Perspectives

*Melanaphis sorghi* is a pest of economic importance in sorghum, oat, and sugarcane crops, which has affected sorghum producers in North America over the past decade. Currently, this pest is managed mainly through the use of chemical insecticides; however, their indiscriminate use has negative effects on health and the environment. This has promoted the study of natural enemies in sorghum fields that have in many cases adapted to the new prey, as well as laboratory research focused on finding microorganisms for biological control to offer alternatives for the management of *M. sorghi*. In the field, natural enemies (predators and parasitoids) and epizootics caused by entomopathogenic fungi have been observed. Evaluations of bacteria against sorghum aphids have demonstrated insecticidal activity on sorghum aphid specimens under controlled conditions. Although most of these alternatives have shown aphidicidal potential against sorghum aphids under some conditions, the actual effectiveness of these biological agents is likely dependent on a number of different abiotic and biotic factors (e.g., soil texture, temperature, humidity, and natural enemies in the field), such that their effectiveness may be highly variable when applied in the field.

The results of the experimental studies reviewed here suggest that despite the limitations of biological control against *M. sorghi*, the use of biological control strategies offers several strong advantages, such as safety and reduced impact on non-target organisms. We know that some biological control products are better than others; however, the use of two or more biologicals against *M. sorghi* should be evaluated, as the combined use of these products could represent a more effective alternative. In response to this pest outbreak, several combination studies of sorghum resistance hybrids and chemical pesticides have been carried out. Several pesticides have suppressed *M. sorghi* while demonstrating low toxicity against natural enemies; this suggests that the combination of these alternatives may be a strategy compatible with integrated pest management programs.

There is a clear need for integrated management programs to establish suitable strategies for the use of biological agents, pesticides, and sorghum resistance hybrids to contribute to the control of sorghum aphid populations in the field in order to minimize risks and maximize benefits.

## Figures and Tables

**Figure 1 plants-13-02873-f001:**
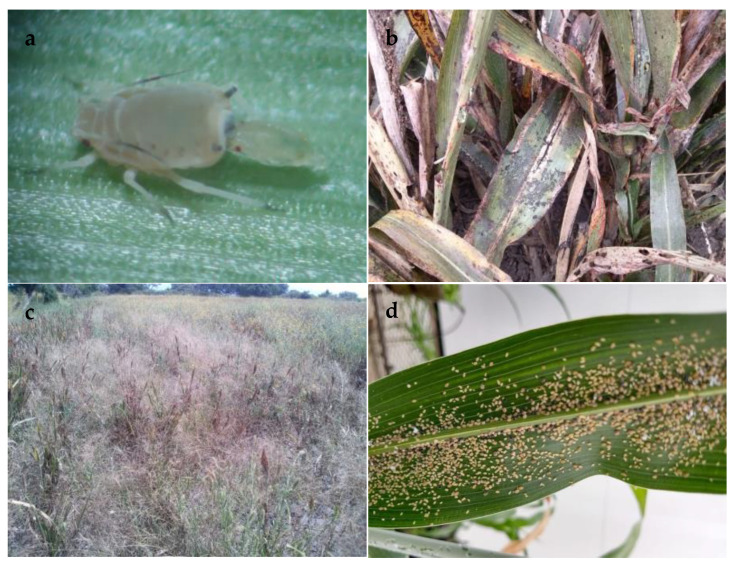
Specimens of *M. sorghi* and the damage they cause to sorghum plants. (**a**) Reproduction through parthenogenesis by a female *M. sorghi*. (**b**) Sorghum plants with accumulation of saprophytic fungi on the leaf surface. (**c**) Total loss of a sorghum cultivar infested by *M. sorhi*; (**d**) Sorghum leaf infested by *M. sorghi*.

**Table 1 plants-13-02873-t001:** Predators reported as natural enemies of *Melanaphis sorghi* in North America and the Caribbean.

Insect	Order: Family	Location	Reference
*Leucopis* sp. Meigen	Diptera: Chamaemyiidae	Sinaloa, Mexico	[a]
*Ceraeochrysa caligata* Banks	Neuroptera: Chrysopidae		
*Ceraeochrysa cubana* Hagen	“	“	
*Ceraeochrysa* sp. *nr. cincta* Schneider	“	“	
*Chrysoperla comanche* Banks	“	“	
*Chrysoperla externa* Hagen	“	“Sinaloa, Chiapas, Guanajuato, and Colima, Mexico. Texas, USA	[a] [b] [f] [g] [e]
*Ceraeochrysa valida* Banks		Sinaloa, Northeastern Tamaulipas, and Colima, Mexico. Texas, USA	[a] [c] [g] [e]
*Chrysoperla carnea* Stephens	“	Sinaloa and Guanajuato, Mexico	[a] [f]
*Chrysoperla rufilabris* Burmeister		Sinaloa and Guanajuato, Mexico. Texas, USA	[a] [f] [e]
*Eupeodes americanus* Wiedemann	Diptera: Syrphidae	Sinaloa, Mexico. Texas, USA	[a] [e]
*Allograpta obliqua* Say	“	Sinaloa, Northeastern Tamaulipas, Guanajuato, and Colima, Mexico. Texas, USA	[a] [c] [f] [g] [e]
*Scymnus sp.* Kugelann	Coleoptera: Coccinellidae	Sinaloa and Southern Tamaulipas, Mexico. Haiti.	[a] [h] [i]
*Cycloneda sanguinea* Linnaeus	“	Sinaloa, Chiapas, Northeastern and Southern Tamaulipas, Colima, Mexico. Texas, USA.	[a] [b] [c] [h] [g] [e]
*Coleomegilla maculata* De Geer		Sinaloa and Northeastern Tamaulipas, Mexico. Texas, USA. Haiti.	[a] [c] [e] [i]
*Hippodamia convergens* Guérin-Méneville	Coleoptera: Coccinellidae	Chiapas, Northeastern and Southern Tamaulipas, Guanajuato, and Colima, Mexico. Texas, USA. Haiti.	[b] [c] [h] [f] [g] [e] [i]
*Olla v-nigrum* Mulsant	Coleoptera: Coccinellidae	Northeastern Tamaulipas, Guanajuato, and Colima, Mexico. Texas. USA. Haiti.	[c] [f] [g] [e] [i]
*Harmonia axyridis* Pallas		Northeastern Tamaulipas and Guanajuato, Mexico. Texas, USA.	[c] [f] [e]
*Coccinella septempunctata* Linnaeus		Northeastern Tamaulipas, Mexico. Texas, USA. Haiti.	[c] [e] [i]
*Diomus terminatus* Say	“	Northeastern Tamaulipas, Mexico	[c]
*Brachiacantha decora* Casey		“	
*Hyperaspis wickhami* Casey		“	
*Collops vittatus* Say	Coleoptera: Melyridae	“	
*Chrysoperla sp.* (*carnea* group) Steinmann	Neuroptera: Chrysopidae	“	
*Diomus roseicollis* Mulsant	Coleoptera: Coccinellidae		[c] [e]
*Scymnus loewii* Mulsant		Northeastern Tamaulipas, and Colima, Mexico Northeastern Tamaulipas, Guanajuato, Colima, Mexico	[c] [f] [g]
*Chilocorus cacti* Linnaeus *Chilocorus stigma* Say	Coleoptera: Coccinellidae	Nuevo Leon, Mexico	[d]
*Leucopis argentata* Heeger	Diptera: Chamaemyiidae	Texas, USA	[e]
*Orius insidiosus* Say	Hemiptera: Anthocoridae	“	
*Chrysopa quadripunctata* Burmeister	Neuroptera: Chrysopidae	“	
*Chrysoperla plorabunda* Fitch	“	“	
*Hemerobius* sp. Linnaeus	Neuroptera: Hemerobiidae	“	
*Pseudodorus clavatus* Fabricius	Diptera: Syrphidae	Texas, USA. Colima, Mexico	[e][g]
*Micromus posticus* Walker	Neuroptera: Hemerobiidae	Guanajuato, Estado de Mexico and Morelos, Mexico	[j]
*Micromus subanticus* Walter	“	Estado de Mexico, Mexico	
*Scymnus dozieri* Gordon	Coleoptera: Coccinellidae	Guanajuato and Colima, Mexico	[f][g]
*Allograpta exotica* Wiedemann	Diptera: Syrphidae		
*Coleomegilla maculata lengi* De Geer	Coleoptera: Coccinellidae	Colima, Mexico	[g]
*Exochomus childreni guexi* LeConte	“	“	
*Hyperaspis* sp. Redtenbacher	“	“	
*Nephus* sp. 1 Mulsant	“	“	
*Nephus* sp. 2. Mulsant	“	“	
*Stethorus* sp. Weise	“	“	
*Ocyptamus antiphates* Walker	Diptera: Syrphidae	“	
*Ocyptamus dimidiatus* Fabricius	“	“	
*Ocyptamus gastrostactus* Wiedemann	“	“	
*Toxomerus maculatus* Macquart	“	“	
*Toxomerus dispar* Fabricius	“	“	
*Toxomerus marginatus* Say	“	“	
*Toxomerus puellus* Hull	“	“	
*Toxomerus politus* Say	“	“	
*Toxomerus pulchellus* Macquart	“	“	
*Toxomerus watsoni* Curran	“	“	
*Orius* sp. Wolff	Hemiptera: Anthocoridae	Southern Tamaulipas, Mexico	[h]
*Eosalpingogaster* sp. Hull	Diptera: Syrphidae	“	
*Eupeodes* sp. Osten-Sacken	“	“	
*Chrysoperla* sp. Steinmann	Neuroptera: Chrysopidae	“	
*Adalia bipunctata* Linnaeus	Coleoptera: Coccinellidae	Haiti	[i]
*Anatis* sp. Linnaeus	“	“	
*Cycloneda sanguinea limbifer* Casey	“	“	
*Episyrphus balteatus* De Geer	Diptera: Syrphidae	“	
*Aphidoletes aphidimyza* Rondani	Diptera: Cecidomyiidae	“	
*Chrysopa oculata* Say	Neuroptera: Chrysopidae	“	

(“) same information as the previous line. References: a = [23], b = [24], c = [25], d = [26], e = [27], f = [28], g = [29], h = [30], i = [31], and j = [21].

**Table 2 plants-13-02873-t002:** Parasitoids reported as natural enemies of *Melanaphis sorghi* in the Americas.

Insect	Order: Family	Location	Reference
*Aphelinus varipes* Foerster	(Hymenoptera: Aphelinidae)	Texas, USA. Coahuila and Southern Tamaulipas, Mexico.	[a] [c] [m]
*Lysiphlebus testaceipes* Cresson	(Hymenoptera: Braconidae)	Texas, Georgia, North Carolina, Alabama and South Carolina, USA. Guanajuato, Coahuila, Northeastern and Southern Tamaulipas, Sinaloa, Colima, Tamaulipas, and Morelos, Mexico	[a] [g] [n] [p] [s] [b] [c] [d] [m] [e] [j] [f] [h]
*Aphidius colemani* Viereck	“	Guanajuato, Mexico	[b] [l]
*Aphidius ervi* Haliday	“	“	
*Binodoxys communis* Gahan	“	“	
*Binodoxys kelloggensis* Pike	“	“	
*Diaretiella rapae* McIntosh	“	“	
*Ephedrus* sp. Haliday	“	“	
*Praon* spp. “	“	“	
*Aphelinus* sp. Dalman	(Hymenoptera: Aphelinidae)	Northeastern Tamaulipas, Mexico	[d]
*Aphelinus nigritus* Howard	“	Texas, Oklahoma, Alabama, and Kansas, USA.	[g] [k] [n] [o] [p] [q]
*Aphidius* sp. Esenbeck	“	Nuevo Leon, Mexico	[i]
*Lysiphlebus fabarum* Marshall	(Hymenoptera: Aphelinidae)	Sinaloa, Mexico	[j]
*Lysiphlebus fritzmulleri* Mackauer	(Hymenoptera: Braconidae)	“	
*Pachyneuron aphidis* Bouché	(Hymenoptera: Pteromalidae)	Southern Tamaulipas, Mexico	[m]
*Pachyneuron muscarum* Linnaeus	“	“	
*Aphelinus mali* Haldeman	(Hymenoptera: Aphelinidae)	“	
*Aphidius platensis* Brèthes	(Hymenoptera: Braconidae)	Minas Gerais, Brazil	[r]

(“) same information as the previous line. References: a = [3], b = [41], c = [42], d = [25], e = [43], f = [44], g = [27], h = [45], i = [46], j = [47], k = [48], l = [28], m = [30], n = [49], o = [50], p = [51], q = [34], r = [4], and s = [52].

**Table 3 plants-13-02873-t003:** Fungi used for the management of *Melanaphis sorghi* in the Americas.

Fungi	Order: Family	Doses	*M. sorghi* Stage	% Mortality (M), Population Density (PD)		Reference
*Beauveria bassiana*	Hypocreales: Cordycipitaceae	200–300 gr 0.7 ha^−1^	Nymphs Adults Winged adults	* control (water) 111.1 ± 37.1 b vs. * 135.2 ± 21.0 e 90.53 ± 16.94 c vs. * 93.8 ± 35.4 d 1.77 ± 4.10 b vs. * 3.8 ± 4.4 d	(PD)	[85]
*Metharizium anisopliae*	Hypocreales: Clavicipitaceae		Nymphs Adults Winged adults	132.3 ± 21.2 d vs. * 135.2 ± 21.0 e 90.8 ± 17.4 c vs. * 93.8 ± 35.4 d 3.1 ± 3.9 c vs. * 3.8 ± 4.4 d	(PD)	
*Beauveria bassiana*	Hypocreales: Cordycipitaceae	101.0 g conidia/L	Nymphs	66.3 ± 11.9	(M)	[86]
Product M.A^®^ 17.5 SP *Beauveria bassiana* + *Metarrhizium anisopliae*	Hypocreales: Cordycipitaceae Hypocreales: Clavicipitaceae	400 g/200 L ha^−1^	Adults Winged adults	1.94 ± 0.10	(PD)	[87]

Means with a different letter are significantly different.

## Data Availability

The original contributions presented in this study are included within the article; further inquiries can be directed at the corresponding author.

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
