# Peer review of "A Review of Biological Control One Decade After the Sorghum Aphid (Melanaphis sorghi) Outbreak"

_plants, 2024, doi:10.3390/plants13202873_

Round 1
Reviewer 1 Report
Comments and Suggestions for Authors
The manuscript with the title “A review of biological control one decade after the sorghum aphid (Melanaphis sorghi) outbreak” provides an interesting review on the natural enemies of M. sorghi in field conditions as well as other biological control alternatives for the control of this pest.
Comment 1
After the brief introduction the manuscript plunges abruptly in biological control. I think it is needed a distinct chapter where the pest species is described including briefly the life cycle, ecological preferences and particularities. Also, since the entire manuscript is dedicated to this pest, the images with it are necessary. Also, description of the damage to plants these pest causes needs to be described and preferably also exemplified with images.
Comment 2
A timeline of the pest causing problems has to be documented and is not enough to simply insist on 2013 in introduction, but possibly provide accurate and well-structure recollection from literature (with sources) where and by who was signaled. For this review dedicated to this pest, documenting an accurate chronologic timeline can picture the trajectory of spreading this pest which is very important in predicting its future trends.
Comment 3
Tables – to me seems unusual to have a distinct column with Taxonomic classifier giving the full name of the taxonomist. As usual practice, full name with authors is needed when lists are presented (such as tables) but it must be given updated to latest acceptable name according to reputable source which shall be mentioned as table footnote (e.g. https://www.gbif.org or another database). The order of columns has to be Order – Family - Species with full name (e.g. Ceraeochrysa caligata Banks).
Comment 4
The mode of action/predation etc. of different control agents has to be briefly explained and presented in a well-organized manner.
Comment 5
Such review works must rely on recent literature, unfortunately old sources are predominant. No sources from last 2 years (none from 2023, 2024 and few from 2022 etc…). Reference list must be updated.
Comment 6
Overall the manuscript has good potential, but it must be better-organized in a more logical sequence, include images with the pest, in order to provide a pleasant experience for the reader.
Best regards.
Comments on the Quality of English Language
English style and grammar improvements are recommended.
Author Response
Reviewer #1
Comment 1
After the brief introduction the manuscript plunges abruptly in biological control. I think it is needed a distinct chapter where the pest species is described including briefly the life cycle, ecological preferences and particularities. Also, since the entire manuscript is dedicated to this pest, the images with it are necessary. Also, description of the damage to plants these pest causes needs to be described and preferably also exemplified with images.
Response: One section where M. sorghi is described was added according to the suggestion of the reviewer. As well as images that describe M. sorghi and the damage caused to sorghum crops were included.
Comment 2
A timeline of the pest causing problems has to be documented and is not enough to simply insist on 2013 in introduction, but possibly provide accurate and well-structure recollection from literature (with sources) where and by who was signaled. For this review dedicated to this pest, documenting an accurate chronologic timeline can picture the trajectory of spreading this pest which is very important in predicting its future trends.
Response: M. sorghi spillover throughout the Americas was included in the section Melanaphis sorghi.
Comment 3
Tables – to me seems unusual to have a distinct column with Taxonomic classifier giving the full name of the taxonomist. As usual practice, full name with authors is needed when lists are presented (such as tables) but it must be given updated to latest acceptable name according to reputable source which shall be mentioned as table footnote (e.g. https://www.gbif.org or another database). The order of columns has to be Order – Family - Species with full name (e.g. Ceraeochrysa caligata Banks).
Response: Columns of the Tables were ordered to be clearer according to the suggestion of the reviewer.
Comment 4
The mode of action/predation etc. of different control agents has to be briefly explained and presented in a well-organized manner.
Response: We showed the use or the natural affectation of M. sorghi populations by around 30 different species of predators and parasitoids, as well as 8 species of bacteria and fungi. Showing their mode of action or predation to each organism represents another review and this is not the aim of this article.
Comment 5
Such review works must rely on recent literature, unfortunately old sources are predominant. No sources from last 2 years (none from 2023, 2024 and few from 2022 etc…). Reference list must be updated.
Response: Current literature concerning the topic of the manuscript was included according to the reviewer’s suggestion.
Comment 6
Overall the manuscript has good potential, but it must be better-organized in a more logical sequence, include images with the pest, in order to provide a pleasant experience for the reader.
Response: We included images of sorghum damage, as well as the pest images.
English style and grammar improvements are recommended.
Response: The English was improved by the American native PhD. Lynna M. Kiere.
Reviewer 2 Report
Comments and Suggestions for Authors
This is an interesting review article that provides a good summary of the management of the sugarcane aphid in North America.
I suggest that the authors reorganize tables 1 and 2, possibly listing each natural enemy once followed by the various studies conducted in different locations.
The references need to be checked carefully, starting with reference 8 I did not see the name of the journals included in the references.
I suggest that the authors consider deleting the word "depredation" possibly using "preyed upon".
Comments on the Quality of English LanguageQuality of English is fine.
Author Response
Reviewer #2
Comment 1
I suggest that the authors reorganize tables 1 and 2, possibly listing each natural enemy once followed by the various studies conducted in different locations.
Response: The Tables 1 and 2 were improved to be clearer.
Comment 2
The references need to be checked carefully, starting with reference 8 I did not see the name of the journals included in the references.
Response: The references were checked according to the reviewer’s suggestion.
Comment 3
I suggest that the authors consider deleting the word "depredation" possibly using "preyed upon".
Response: The word "depredation” was changed to “preyed upon” in the manuscript according to the reviewer´s suggestion.
Reviewer 3 Report
Comments and Suggestions for Authors
1. The title is clear but could be more specific regarding the focus on biological control agents, particularly in relation to integrated pest management (IPM). Consider adding a reference to IPM to attract a reader.
2. The abstract succinctly summarizes the review but lacks a detailed mention of the significant challenges in implementing biological control and IPM strategies. Briefly addressing the limitations of biological control (e.g., environmental factors, variable success rates) would provide a more balanced overview.
3 The introduction effectively highlights the emergence of M. sorghi as a major pest in North America. However, some parts are repetitive, particularly in the explanation of the pest’s impact on sorghum crops. Streamlining this section could improve readability. The introduction should also better contextualize the review by briefly explaining the importance of IPM and how biological control fits into this broader strategy. While this is discussed later, providing context early on would enhance the narrative.
4. The methodology section is not clearly defined. While this is a review paper, it is essential to explain the criteria used for selecting studies and the sources of data. Were there any specific databases or journals targeted for the literature review? A brief explanation of the search strategy would improve transparency.
5. Predators., This section is informative but could benefit from a clearer distinction between predator species that are more effective at different life stages of the aphid. More details on how environmental factors, such as temperature and landscape, impact predator efficiency should be included earlier
6. Parasitoids., This section is well-organized and provides a good summary of the role of parasitoids. However, the discussion on hyperparasitism should be expanded, particularly regarding its impact on the long-term success of parasitoid-based biological control.
7. The paper effectively discusses natural enemies but would benefit from a more in-depth analysis of how these biocontrol agents fit into a comprehensive IPM strategy. Highlighting the synergy between biological controls, resistant cultivars, and judicious pesticide use would offer practical recommendations for farmers and pest managers.
8. The section on environmental and landscape influences is insightful. It would be valuable to provide more detailed examples of how these factors specifically affect predator and parasitoid populations, possibly drawing on additional case studies.
9. The review offers a balanced discussion on the impact of pesticides on natural enemies. However, some conflicting data are presented without sufficient explanation (e.g., conflicting effects of sulfoxaflor). Clarifying these inconsistencies or providing more detailed explanations for them would improve the reader’s understanding of pesticide impacts.
10. The section on biopesticides, especially entomopathogenic fungi and bacteria, is valuable but could benefit from more specific examples of field applications and outcomes. Additionally, a comparison of biopesticide efficacy to chemical pesticides would enhance the discussion on their practical use in IPM.
11. The tables are informative and comprehensive, particularly those summarizing predator and parasitoid species. However, some tables contain a large amount of information, which can make them difficult to interpret quickly. Consider reorganizing long tables (e.g., Table 1 and Table 2) into more digestible segments or using color bands to differentiate between groups of data (e.g., grouping species by location or type of predator/parasitoid).
12. The conclusion effectively summarizes the potential of biological control but should better address the future research needs and challenges in implementing these strategies on a larger scale. A more detailed discussion on the importance of monitoring and adapting biocontrol strategies over time would add depth.
Author Response
Comment 1. The title is clear but could be more specific regarding the focus on biological control agents, particularly in relation to integrated pest management (IPM). Consider adding a reference to IPM to attract a reader.
Response: Integrated pest management is focused on the implementation of a set of methods (chemical, plant breeding, conservation of entomofauna, biological control, among others) to keep under the economical threshold of insect pests, on the other hand, our review is focusing particularly on biological control.
Comment 2. The abstract succinctly summarizes the review but lacks a detailed mention of the significant challenges in implementing biological control and IPM strategies. Briefly addressing the limitations of biological control (e.g., environmental factors, variable success rates) would provide a more balanced overview.
Response: The abstract was improved according to the challenges that face the effectiveness of biological control agents in the field.
Comment 3. The introduction effectively highlights the emergence of M. sorghi as a major pest in North America. However, some parts are repetitive, particularly in the explanation of the pest’s impact on sorghum crops. Streamlining this section could improve readability. The introduction should also better contextualize the review by briefly explaining the importance of IPM and how biological control fits into this broader strategy. While this is discussed later, providing context early on would enhance the narrative.
Response: The objective of this manuscript is to show the alternatives of biological control addressed in the management of M. sorghi, and speak about IPM exceeds the aim of this manuscript. IPM is a very important science but is a very complex discipline which with the information addressed in this manuscript we cannot cover.
Comment 4. The methodology section is not clearly defined. While this is a review paper, it is essential to explain the criteria used for selecting studies and the sources of data. Were there any specific databases or journals targeted for the literature review? A brief explanation of the search strategy would improve transparency.
Response: The methodology used to do the review was included according to the reviewer’s suggestion. Line 110 to 114.
Comment 5. Predators., This section is informative but could benefit from a clearer distinction between predator species that are more effective at different life stages of the aphid. More details on how environmental factors, such as temperature and landscape, impact predator efficiency should be included earlier.
Response: In the Line 180 we mentioned that we address this potential interactive effect in more detail in the section “Natural enemies of M. sorghi mediated by landscape and weather”.
Comment 6. Parasitoids., This section is well-organized and provides a good summary of the role of parasitoids. However, the discussion on hyperparasitism should be expanded, particularly regarding its impact on the long-term success of parasitoid-based biological control.
Response: We discussed briefly about hyperparasitism problematic in the success of the classical biological control. Line 279-283.
Comment 7. The paper effectively discusses natural enemies but would benefit from a more in-depth analysis of how these biocontrol agents fit into a comprehensive IPM strategy. Highlighting the synergy between biological controls, resistant cultivars, and judicious pesticide use would offer practical recommendations for farmers and pest managers.
Response: Through the manuscript we mention the importance of biological control and its application in integrated management programs, however, we cannot deepen into the IPM, because it is out of the aim of this review of biological control advances in the management of M. sorghi. Line 34 abstract, line 375-377, line 391-394, line 490-492, line 493-496.
Comment 8. The section on environmental and landscape influences is insightful. It would be valuable to provide more detailed examples of how these factors specifically affect predator and parasitoid populations, possibly drawing on additional case studies.
Response: Line 362-369 provide more detail about environmental and landscape influences on the entomofauna according to the reviewer´s suggestion.
Comment 9. The review offers a balanced discussion on the impact of pesticides on natural enemies. However, some conflicting data are presented without sufficient explanation (e.g., conflicting effects of sulfoxaflor). Clarifying these inconsistencies or providing more detailed explanations for them would improve the reader’s understanding of pesticide impacts.
Response: Lines 377-380 explain clearer the affectation of parasitoids due to the application of sulfoxaflor.
Comment 10. The section on biopesticides, especially entomopathogenic fungi and bacteria, is valuable but could benefit from more specific examples of field applications and outcomes. Additionally, a comparison of biopesticide efficacy to chemical pesticides would enhance the discussion on their practical use in IPM.
Response: This is not a review of IPM, We cannot address in a deeper way the use of biological agents together with the IPM because will lose the aim of this review. We cannot compare the chemical pesticides and biopesticides, biological agents need to carry out an infection process, while pesticides act quickly on the nervous central system.
Comment 11. The tables are informative and comprehensive, particularly those summarizing predator and parasitoid species. However, some tables contain a large amount of information, which can make them difficult to interpret quickly. Consider reorganizing long tables (e.g., Table 1 and Table 2) into more digestible segments or using color bands to differentiate between groups of data (e.g., grouping species by location or type of predator/parasitoid).
Response: Columns of the Tables were ordered to be clearer.
Comment 12. The conclusion effectively summarizes the potential of biological control but should better address the future research needs and challenges in implementing these strategies on a larger scale. A more detailed discussion on the importance of monitoring and adapting biocontrol strategies over time would add depth.
Response: We addressed in lines 481 to 489 the importance of these studies, as well as the need to apply the use of biological control agents together with plant breeding, and chemical products, and also propose the IPM programs in focus on the sorghum aphid.
Round 2
Reviewer 1 Report
Comments and Suggestions for Authors
Dear authors,
the main issues were addressed in this revised version.
Best regards.
Author Response
Comments 1.
the main issues were addressed in this revised version.
Response: Dear reviewer, thank you for your suggestions and comments to improve this review.
Grammar and style of English were improved by American native.

Reviewer 2 Report
Comments and Suggestions for Authors
I suggest that the authors clearly state that publications, prior to the Nibouche et al 2021 study that demonstrated the difference between M. sacchari and M. sorghi, were likely examining interactions between natural enemies and M. sorghi. Currently, when looking at papers cited in references, many authors present results on the interactions of natural enemies with M. sacchari. The authors need to clarify this issue for readers.
Tables 1 and 2 need to be reformatted. Each natural enemy should be listed once, not multiple times. It is very difficult to read these tables in their current format. I suggest that the importance of these two tables is to summarize the presence of natural enemies in multiple locations. Thus, the suggestion to list each natural enemy only once, followed by locations and references for each location.
Author Response
Comments 1.
I suggest that the authors clearly state that publications, prior to the Nibouche et al 2021 study that demonstrated the difference between M. sacchari and M. sorghi, were likely examining interactions between natural enemies and M. sorghi. Currently, when looking at papers cited in references, many authors present results on the interactions of natural enemies with M. sacchari. The authors need to clarify this issue for readers.
Response: The M. sacchari literature generated before the Nibouche study was clarified for the readers according to the reviewer´s suggestions. Lines 53 to 57.
Comments 2.
Tables 1 and 2 need to be reformatted. Each natural enemy should be listed once, not multiple times. It is very difficult to read these tables in their current format. I suggest that the importance of these two tables is to summarize the presence of natural enemies in multiple locations. Thus, the suggestion to list each natural enemy only once, followed by locations and references for each location.
Response: The Tables were improved according to the suggestion of the reviewer.
Grammar and style of English were improved by American native.

Reviewer 3 Report
Comments and Suggestions for Authors
The manuscript was revised dramatically now accept it in present form .
Comments on the Quality of English LanguageMinor editing of English language required.
Author Response
Comments 1.
The manuscript was revised dramatically now accept it in present form.
Response:
Dear reviewer, thank you for your suggestions and comments to improve this review.
Grammar and style of English were improved by American native.
